# Barriers and facilitators to oral pre-exposure prophylaxis uptake among adolescents girls and young women at elevated risk of HIV acquisition in Lilongwe, Malawi: A qualitative study

Simon C. Nicholas[1,2]*, Maureen Matewere[3], Agatha Bula[3], Mercy Tsidya[3], Mina C. Hosseinipour[3,4], Mitch Matoga[3], Alinane Linda Nyondo Mipando[1,5]

1 Department of Health Systems and Policy, Kamuzu University of Health Sciences, Blantyre, Malawi, 2 Department of Pharmacy, University of North Carolina Project, Lilongwe, Malawi, 3 University of North Carolina Project, Lilongwe, Malawi, 4 Department of Medicine, University of North Carolina at Chapel Hill School of Medicine, Chapel Hill, North Carolina, United states of America, 5 Department of Womens' and Childrens' Health, University of Liverpool, Liverpool, United Kingdom

* snicholas@unclilongwe.org

## Abstract

Among the estimated 12,500 new HIV infections in Malawi among people aged 15-24 each year, 70 percent occur in Adolescent Girls and Young Women (AGYW). The Ministry of Health (MoH) in Malawi rolled out an oral Pre-Exposure Prophylaxis (PrEP) prevention program targeting populations at elevated risk of HIV acquisition, including AGYW, in 2021. Since PrEP roll-out, there has been limited research exploring the factors that influence uptake of PrEP among AGYW. This study explored the barriers and facilitators to the uptake of PrEP among AGYW at elevated risk of HIV acquisition. it was an exploratory qualitative study conducted at Kawale Health Center in Lilongwe, Malawi, in February 2023, which employed a phenomenological design. Data were collected using semi-structured in-depth interviews and vignettes from purposively sampled 20 AGYW and 10 health care workers (HCWs) based on their PrEP status (on PrEP versus not on PrEP) and involvement in PrEP provision, respectively. The data were digitally recorded, managed using NVivo software and analysed using a thematic approach guided by the Consolidated Framework for Implementation Research (CFIR). AGYW identified perceived HIV risk and vulnerability and PrEP knowledge as facilitators. HCWs identified AGYW perceived HIV risk, HCW altitudes, and availability of youth friendly service center and resources as facilitators to PrEP uptake. Barriers identified by AGYW included PrEP side effects, limited PrEP information, lack of privacy, stigma, and lack of transportation. HCWs identified limited resources and burden of work as barriers. In conclusion, PrEP's full potential as an HIV prevention tool for AGYW requires a holistic approach that considers their particular requirements, removes systemic hurdles, and guarantees access to high-quality services. In addition, there is a need to create demand to increase the uptake of PrEP.

**Data availability statement:** All relevant data are within the paper and its Supporting Information.

**Funding:** SCN was supported by Malawi HIV Implementation Research Scientist Training (MHIRST) program under the NIH Fogarty grant # 2D43 TW010060. This grant was for his scholarship for the Masters of Global Health Implementation.

**Competing interests:** LNM is an Academic Editor for PLoS one and PLoS Global Public Health. These competing interests will not alter adherence to PLOS Global Public Health policies on sharing data and materials. All other authors declare that no competing interests exist.

## Introduction

Globally, Adolescent Girls and Young Women (AGYW) aged 15–24 are more vulnerable to HIV than other demographic groups, with over 1000 AGYW acquiring HIV daily [1]. Similarly, in sub-Saharan Africa (SSA), AGYW are disproportionately affected by the HIV epidemic, accounting for 25% of all newly acquired HIV infections [2]. Compared to their male counterparts, AGYW are 2.5 times more likely to acquire HIV infection through heterosexual sexual contact [3]. In Malawi, 12,500 newly acquired HIV infection occur among young people aged 15–24 each year, and 70 percent of these are AGYW [4]. Since 2018, the HIV prevalence in AGYW in Malawi has stagnated at around 5%, while there has been a decrease from 1.9% to 1% among adolescent boys and young men [5].

The disparities in the impact of HIV among AGYW are attributed to high-risk behaviour and sociodemographic and socioeconomic factors [6]. High risk behaviour associated with HIV among AGYW include early sexual debut, multiple sexual partnerships, limited condom usage, intimate partner violence, and intergenerational and transactional sex [6]. Furthermore, other factors such as age, marital status, low level of education, lack of employment, and place of residence have also been associated with the risk of HIV among young people [7]. A study in Malawi showed that people residing in urban areas had a 2.2 times greater risk of being infected with HIV compared to their counterparts in the rural areas [8]. Unfortunately, AGYW are often left out of HIV prevention and treatment services [9]. While effective strategies for HIV prevention exist, such as condom use and male circumcision, they remain non-gender sensitive.

Although female condoms are available, the use of condoms remains predominantly male-controlled, which restricts women's capacity to utilize them as they must negotiate their use with sexual partners [10]. For women with less influence in relationships, this presents difficulties. Research indicates that women who have experienced intimate partner violence are much less confident when negotiating condom use with a partner, increasing their chance of contracting HIV [11]. Compounded with other multiple sexual and reproductive health barriers such as persistent hesitancy and shyness, negative beliefs and attitudes associating family planning services, condom access and usage is difficult among AGYW [12].

Another HIV prevention intervention is oral PrEP, which entails the use of antiretroviral (ARV) medications by HIV-negative persons to prevent HIV acquisition. Oral PrEP was recommended by the World Health Organization (WHO) as a prevention option for persons at high risk of HIV acquisition in 2015 [10], and is now widely available. Malawi Ministry of Health (MoH) implemented it in 2021 [13].

Although PrEP has shown high efficacy in clinical trials, uptake of PrEP by AGYW across the region has been slow [14–17]. Notably, there has been no research to date detailing the factors that influence the uptake of PrEP among AGYW since it was rolled out in 2021. This study explored the barriers and facilitators to oral PrEP uptake among AGYW in Lilongwe, Malawi.

## Methods

### Study design

This was an exploratory qualitative approach using a Phenomenological design. Data were collected from 1 February 2023 to 29 February 2023. We used this design to unravel AGYWs experiences with and provide a better understanding of the PrEP program in Malawi. Data were collected using vignettes and in-depth interviews among AGYW and in-depth interviews only among HCWs who provided PrEP at the facility. We used vignettes with a short story about hypothetical circumstances around PrEP access and usage among AGYW.

## Study setting

This study was conducted at Kawale Health Center in Lilongwe, Malawi. Kawale Health Center is one of the busiest urban health facilities under Lilongwe District Health Office. The services offered at the facility are HIV testing services, youth friendly health services, HIV pre-exposure prophylaxis (PrEP), sexually transmitted infections (STI) treatment and prevention, Voluntary Medical Male Circumcision (VMMC), and anti-retroviral therapy (ART). HIV Testing Service (HTS) and ART at this clinic are offered by Lighthouse, a center of excellence in HIV care, in conjunction with the Ministry of Health [18]. Kawale health center serves a population of about 326,900. The facility had 10 HCWs trained in PrEP services including clinicians, medical assistants, nurses and HTS counsellors. The facility had enrolled 140 women on PrEP around the implementation period. Of the women initiated on PrEP, 28 were adolescents and young women, representing 20%.

## Sampling and recruitment of study participants

We selected our study participants from AGYW accessing youth-friendly health services at Kawale health centre. Purposive sampling technique was used to select 20 AGYW based on their PrEP status [19]. This study deliberately chose non-probability sampling [20] to include both AGYW who accepted PrEP and those that were not on PrEP (10 AGYW in each group). By using this sampling method, there was an assurance that the sample represents client perceptions as well as provided the chance to learn from two important extreme groups on PrEP use. Ten (10) Health Care workers were also purposively selected based on their involvement in the provision of PrEP services at the facility. We included those that had provided PrEP services for more than 6 months at the site and were available and willing to participate in the study. Participants were approached at the facility by the investigators and the PrEP focal person of the facility and were briefed of the study. They were interviewed at their convenience upon agreement to participate in the study.

## Data collection

Data were collected from 1st February to 28th February 2023 by a well-trained female research assistant and the Principal Investigator. The participants were asked to sign on a written informed consent form upon agreeing to participate in the study. Following written informed consent, the investigators utilized the vignettes to share a story to AGYW followed by a semi-structured guide which was administered either in English or Chichewa (national language) based on preference. The vignettes with interview guide (S1 Appendix) for AGYW included the short story with questions about PrEP knowledge, facilitators and barriers to PrEP uptake guided by the Consolidated Framework for Implementation Research (CFIR). HCWs were interviewed using IDI guide which had open-ended questions (S2 Appendix), also, it included open ended questions about access to training on PrEP guidelines, knowledge of PrEP and barriers and facilitators to PrEP program implementation among AGYW.

The interviews were conducted face-to-face and were audio-recorded in both English and Chichewa language to capture every detail. The Chichewa interviews were translated and transcribed to English verbatim to avoid loss of information. Interviewing participants after accessing services and health care workers at their convenient time was intended to ensure undivided attention during interviews. The transcripts were assigned unique numbers for confidentiality and easy record follow-up. The transcripts, recorders and consent forms were kept in a lockable cabinet and was only accessed by the investigators. The electronic records were stored on a password protected computer with strict access by investigators. The interviews, on average, lasted 21 minutes, ranging from 17 to 33 minutes.

### Data analysis

The transcripts were analysed using a thematic analysis approach as suggested by Braun and Clarke [21]. Data were managed using qualitative software NVivo.

SN and ALNM read initial transcripts independently, repeatedly, to familiarize themselves with the data and at the third round of reading they independently coded the transcripts to develop a code book. The coding process followed deductive and inductive approaches. We use the CFIR framework to deduce the codes while the inductive codes were realised from the data. After independently coding the data, SN and ALNM discussed the initial codes that were developed and discussed areas where they differed until resolution. SN developed the entire codebook that was checked by ALNM and by AKB. SN then coded the entire dataset using the codebook while also adding more inductive codes as realised from the data. transcripts several times while listening to the audios in order to become completely immersed in the facts and guarantee proper transcription. Any interesting ideas and patterns the data may reveal will also be noted by the researchers. At the time of coding SN had periodic meetings with ALNM and AKB to oversee the progress and discuss other emerging issues or codes. After all the transcripts were coded, we combined related and comparable codes under an overarching theme that were either deduced from the conceptual framework or realised from the data. We reviewed the themes realised multiple times to ensure that they are representative of the data and were coherent. The matrix that compared the conceptual framework and the themes realised is shown in Table 1. The quotes attached to the themes were thoroughly reviewed to ensure that they were representative and that there was no overlap. The final themes were then reviewed by all researchers to ensure that they were correct and complete.

### Ethical approval and consent to participate

An ethical approval to conduct the study was given from the College of Medicine Research Ethics Committee (COMREC), certificate number P.11/22/3854, and permission was obtained from Lilongwe District Health Office Additionally, informed consent was sought from all participants before participating in the study. Interviews were conducted in a private room and no identifiable information was used to ensure privacy and confidentiality.

## Results

### Demographic characteristics of AGYW

Out of the 20 AGYW recruited, median age was 23 (IQR:21, 24) and most 16 (80%) were single and had up to secondary level education.

### Demographic characteristics of health care workers

Of the 10 recruited HCWs, 8 (80%) were females, median age was 38.5 years (IQR: 34, 40) the mean years of professional experience of the HCWs was 10. Among them, three were clinical officers, two were senior medical assistants and two were medical assistants. These were involved in the screening of clients and prescribing of Prep. There were also two nurses who were responsible for screening of clients and do PrEP sensitisation talks to clients. The last one was HTS counsellor who was responsible for HIV counselling.

### Barriers and Facilitators to PrEP uptake among high risk AGYWs

A CFIR domain-by-domain analysis of PrEP uptake among AGYWs revealed barriers and facilitators (Table 1). The AGYW contributed participant characteristics, intervention characteristics, and inner and outer setting perspectives, while the HCW contributed inner and outer setting perspectives, intervention characteristics, and implementation process perspectives.

**Table 1. Summary of barriers and facilitators based on CFIR domains and associated constructs.**

| CFIR domains. | Facilitators | Barriers |
|---|---|---|
| **Intervention Characteristics:** Aspects of an intervention that may impact implementation success (adaptability and complexity) [22]. | | **Complexity:** **Adherence Problems due to PrEP attributes** The side effects of PrEP were a barrier to medication uptake. |
| **Characteristics of Individuals:** Individuals' beliefs, knowledge and personal attributes that may affect implementation of people implementing or receiving the intervention [22]. | **Knowledge of PrEP:** Adolescent girls and young women and health care providers, recognized the value of PrEP. **Vulnerability and perceived risk of HIV infection.** Health care workers and AGYWs perceived PrEP to be useful because of a 'high-risk' perception among AGYW because of their increased exposure to HIV and vulnerability. | **Limited knowledge about PrEP.** Some AGYWs would not want to use PrEP as they consider PrEP to be ARTs used by the people living with HIV. |
| **Inner Setting:** Characteristics of the implementing organization such as team culture, relative priority of the intervention, leadership engagement, and the compatibility of the intervention with the organization [22]. | **Availability of youth friendly centre** AGYWs expressed that good attitude of providers at Kawale health Centre and availability of youth friendly health services enhances successful PrEP uptake among them. **Availability of resources:** Availability of PrEP, and other resources such as HIV test kits, were described as factors that makes it easy for PrEP to be provided. | **Burden of work** Too much workload on the side of providers and shortage of trained PrEP providers. **Lack of privacy** The study showed that the facility lacks space that can provide confidentiality to AGYWs and this compromises uptake as some of them would not want to be seen getting PrEP by other people. |
| **Outer Setting:** External influences on intervention implementation including patient needs and resources, external policies and incentives, community culture and attitudes [22]. | **Patient needs and resources:** The policy on continuous sensitisation in all PrEP-providing facilities by MoH helped to improve understanding of the benefits of PrEP. | **Patient needs and resources:** **Stigma** Most of the adolescents perceived and anticipated stigma from parents, boyfriends and the community that may hinder AGYWs from taking PrEP. **Limited resources:** Shortage of supplies and other resources at the facility, e.g., gloves and lack of transport to come for PrEP refill. |
| **Process:** Strategies used, the presence of key intervention stakeholders and influencers including opinion leaders, stakeholder engagement, and project champions [22]. | **Engaging:** **Availability of key intervention stakeholders** The facility engaged Light House health care providers to help facilitate PrEP implementation. The staff work hand in hand in the provision of PrEP at the facility. | |

## Facilitators to PrEP uptake

### Characteristics of individuals

**Knowledge of PrEP and its benefits.** Knowledge of PrEP was a facilitator for both AGYW and HCWs. The knowledge included about how PrEP is administered and how it works, its side effects, adherence, and its effectiveness in HIV prevention. AGYW stated that understanding PrEP increased their confidence in continuing the medication and continuously reinforced their reasons for taking it.

*"When I realized that my husband takes ART drugs, it's when I decided that I should protect myself, what can I do, so when I came to explain it to the provider, it's when he/she told me that there are some drugs that can protect me, so it's when I told them that it's good that I should start taking them that I should not contract the virus." AGYW-05*

**Vulnerability and perceived HIV infection risk.**  Both AGYW and HCWs perceived PrEP as useful for HIV negative AGYWs. Participants reported that AGYWs usually have multiple sexual partners, including older men. Additionally, there was a perception that AGYWs are more likely to engage in risky sexual behaviors, such as exchanging sex for money or gifts, which can further increase their risk of HIV.

*"I think it is a good idea to give the adolescent girls and young women because most of the adolescents usually have unprotected sexual intercourse because in most cases, is unprepared for. They are the people at high risk because they have multiple sexual partners and there is usually what they call intergenerational sex which involves people of different generations having sex and these girls don't have any say when they are having sex with a man aged 50 or 60 because of power difference. So, it is a good idea to give them PrEP so that they are protected from HIV." HCW-10*

*"What motivated me as a sex worker to be taking PrEP is that sometimes we are forced by some men to have unprotected sex. We rush to the hospitals to get PEP when this happens because we feel that if a man forces a sex worker to have unprotected sex it means he is positive. In my case when a man does this to me, I show him the bottle of PrEP and also encourage him to go and get their own bottle." AGYW-16*

## Inner setting

**Attitude of HCWs and availability of the youth friendly service centre.**  HCWs and AGYW stated that a positive attitude of providers at Kawale Health Centre, such as welcoming and fast-tracking PrEP users (AGYW) facilitated PrEP uptake. Health care workers stated that the availability of a youth friendly facility at the clinic makes it easier for the AGYW to access PrEP.

*"In terms of provider attitude, I have never heard anyone complain about provider attitudes because even if the clinic is full, when we have a PrEP client, we fast track them so that they should not take long in the clinic." HCW-07*

*"They welcome us well. And another thing that encourages us is the way the providers that we meet encourage us." AGYW-05*

*"The way they welcome me here at Kawale and the counselling that I get here at Kawale is what influences me to come here." AGYW-06*

*" I think the availability of the youth corner becomes one of the areas which facilitates the youths getting services here. The availability of their peers, the availability of trained providers and sensitization which provides knowledge on the need for them to demand for services at this facility." HCW-10*

**Availability of resources.**  HCWs argued that the availability of PrEP, as well as other resources such as HIV test kits, makes it easier for AGYWs to receive PrEP and benefit from it. Participants were encouraged with the availability of PrEP drugs at the facility and they knew that the drugs are available at all times.

*"The availability of drugs is another factor that enable us provide PrEP to those in need."* HCW-06

**Availability of PrEP focal point person.** The HCWs also stated that the availability of a PrEP focal person who simultaneously acts as a mentor facilitated PrEP provision. Having someone dedicated to providing PrEP care and support helped HCWs to become knowledgeable about PrEP and would be able to answer any questions AGYW may have. This helped to increase AGYW confidence in the PrEP and ensured a seamless PrEP provision process.

*"The focal person is responsible for ensuring that PrEP services are provided at all times. The focal person ensures that all the data is written, documentation is done and the issue of availability of PrEP as well. There is also a mentor for cadres that are just coming in"* HCW-10

## Barriers to PrEP implementation and uptake

### Intervention characteristics

**Adherence problems due to PrEP side effects.** The side effects of PrEP medications remain a concern among PrEP beneficiaries. The side effects were mentioned as barriers by both AGYW that are on PrEP and those that are not on PrEP. Hence the side effects were barriers to PrEP uptake and PrEP continuation. HCWs reported that AGYWs' fear of PrEP side effects may lead to poor PrEP adherence and its usage that may result to drug resistance.. Abuse of PrEP was also highlighted as some AGYW would come to the clinic to get PrEP not because they wanted to use it but as a peer pressure.. In addition, AGYW may abuse PrEP by sharing with friends without prescription leading to inadequate dosing and drug resistance..

*"Maybe, I can talk of the adherence challenges; they may be experiencing the adherence challenges. I know before prescribing PrEP we are supposed to conduct health education but we know that PrEP is going to be taken for so long. So, sometimes we can have that challenge of adherence."* HCW-03

*"For every drug, even the usual drugs that we take, if you have never taken such drugs or if your body is not used to the drugs, some people develop rash, some have loss of appetite… different side effects. But for PrEP… we are given for 30 days and if you take for one week, you still have the side effects but when you continue taking, the side effects stop. Like in my case, it took two weeks for the side effects to resolve. However, they are good drugs and they prevent from HIV."* AGYW-12

### Characteristics of individuals

**Lack of PrEP awareness.** It was noted from both AGYWs and HCWs that some AGYWs may not want to use PrEP because of lack of information such as length of period for taking PrEP and side effects.

*"What can infringe me is that… of course I just heard in bits, I am not quite sure… So maybe if there could be someone that can explain to me very well about PrEP, then I can have some idea then maybe I can take part in it, but because of lacking the knowledge, it's difficult for me to take them."* AGYW-04

*"People say PrEP is not good, like for those that once used it say PrEP makes you sick, it gives you headaches, dizziness and others. They say that when you take it at night you get bad*

*dreams. With that in mind, the youth get scared to take PrEP. Others also don't understand what PrEP is, they think that you are HIV positive and that you are just hiding, they are ARVs and not PrEP."* AGYW-03

## Inner setting

**Burden of work.** HCWs noted that workload on the side of providers and shortage of qualified PrEP providers hindered quality PrEP provision to the AGYW. It was noted that the clinician who attended to the clients at the OPD was also responsible for the dispensation of PrEP. The participants reiterated that in an environment that already suffers from staff shortages, the incorporation of PrEP as an additional clinic service was a challenge s. They stated that the providers are already responsible for OPD services such that adding the role of being PrEP providers increases their scope of work.

*"I will talk about the workload of clinicians at OPD. I have already said that the registration, the enrolment, is usually done at OPD where there is usually one clinician, who is attending to all the OPD clients, and these adolescents who are coming to seek PrEP services. So, I would say, much attention is given to those who have come because they are sick.* HCW-01

*"Yes, this facility can provide PrEP with no challenges, however we lack some resources such as well-trained providers. There are just very few and inadequate space to carry out services because right now we do that at OPD. So, at OPD it is only us who carry out the services because the nurses don't have space in the said OPD."* HCW-04

**Lack of privacy.** Although AGYW were fast-tracked through the clinic, HCWs and AGYW stated that client flow at the clinic impeded access to PrEP because of limited private space for PrEP service provision which compromised their privacy. In some cases, some AGYW went back home without PrEP because they were tired of long waiting time.

*"Factors such as lack of adequate space as some patients feel shy like to come to OPD. It becomes difficult for the patient to get drugs after seeing a lot of other patients, in so doing the patient may opt to just going back home without getting the drugs. Or sometimes the patient might be in the queue and upon seeing that it will take time to be assisted s/he just leave the line for home. Sometimes the clients have a feeling that if they keep on coming to get the drugs, some will view them as people who love sexual activities."* HCW-04

**Limited resources.** Some HCWs explained that despite having most resources available, they do not conduct some critical tests at the facility due to lack of resources. They gave an example of a creatinine clearance test which is one of the indicators of PrEP eligibility. The HCWs reported that the only way to get the test done was to refer the patient to a tertiary hospital which was not always feasible. The participants stated that they preferred to receive PrEP services at a single facility than being referred to other facilities for other tests which becomes a deterrent.

*"As a facility, we only do Hepatitis and HIV tests but we don't do creatinine clearance or kidney screening. I would like these important tests to be done here as they are indicators for PrEP eligibility."* HCW-06

Some AGYWs and HCWs stated that some AGYWs had problems getting PrEP refills because of transportation challenges. Participants gave examples of coming far from

the facility which affected their mobility and access to the facility. They end up either missing their appointments or delay getting refills, which can negatively affect their adherence.

*"In terms of transport, it happens that maybe the client did not come on her/his scheduled visit… when we ask them 'why didn't you come on your scheduled visit date'? That is when they tell us about issues of transport. So, transport is really a barrier."* HCW-07

*"Sometimes it could be that you stay far from the facility, so transportation can hinder you from going to receive the drugs."* AGYW-05

## Outer setting

**Stigma.** The majority of AGYW experienced and anticipated stigma from parents, lovers, and the community as key obstacles that prevented them from using PrEP, even though they were aware of its goals and advantages. According to the participants, PrEP shares the stigma associated with antiretroviral medication (ART) in the community due to its identical packaging and look. The use of PrEP is linked to increasing promiscuity, commercial sex workers, and HIV-positive individuals, according to the respondents.

*"Firstly, most youths are afraid to get PrEP because with PrEP you take the medicine daily so others are like; 'Should I be going to the hospital every time I need PrEP? My friends will be laughing at me'. Others are afraid of their friends and also their parents. If parents realize that my child is taking PrEP, they may have all sorts of ideas. Also, I think religion; depending on the religion where one comes from, the young person may be afraid to say 'I am from this religion and I am taking PrEP it will be viewed as something else'. Also, culture: other cultures inhibit the youth from taking PrEP."* AGYW-01

*"……. This is because PrEP is distributed at ART by Lighthouse. So, when people see you going there, they feel that you are going to receive ART. So, she was concerned because she is going there to get PrEP for HIV prevention while other people are going to the same place to get ARVs"* AGYW-08

## Discussion

Health care workers and adolescent girls and young women were interviewed in-depth to identify key facilitators and barriers to PrEP uptake among AGYW. Among the facilitators were AGYW perceptions of vulnerability and risk of HIV, knowledge of PrEP, and availability of youth-friendly service centers. The barriers to PrEP use included side effects, burden of work, insufficient privacy, and the stigma associated with PrEP.

According to our study, AGYWs' perceptions of risk and vulnerability contributed to their uptake of PrEP. It was noted by participants that they were willing to use PrEP since they were HIV-negative and believed that the use of PrEP would prevent HIV. This is consistent with previous findings in a study where PrEP was integrated into a family planning clinic in Kenya [23]. Additionally, it is believed that risky sexual activities and self-perceived risk are some of the variables influencing the use of PrEP in the US [24]. According to this study, AGYW begin PrEP to guard against contracting HIV from their partners. As with a research conducted in Ghana, PrEP was approved because it offered an extra degree of HIV protection [25].

As a preventive option, PrEP should be known to individuals at risk and healthcare providers before any other factors start to influence uptake. In addition to being empowered by understanding PrEP's benefits as a HIV prevention method, AGYW also disclosed the use

of PrEP to their male sexual partners, thereby fostering the use of PrEP among men. Among the AGYW participants that were using PrEP, six had sex work experience that made them vulnerable to HIV infection. We noted that the participants who were on PrEP were encouraged by their exposure to HIV than those who were not on PrEP. Among the participants who had sex work experience, one admitted to disclosing her PrEP status to her clients, which then led to her partner initiating PrEP. PrEP status disclosure to the partner can reduce stigma and improve PrEP adherence as people are more likely to continue taking PrEP if they have their partners' support [26,27].

Similarly, other researchers have reported high PrEP uptake following PrEP awareness among AGYW highlighting the need for awareness creation [28]. According to our research, the majority of AGYW had unfavorable encounters with peers using PrEP and were either unaware of it or had been misinformed about it. The participants suggested that the general population be provided with accurate information regarding PrEP's effectiveness and side effects. This will assist AGYW in getting the information they need to decide whether to use PrEP. A study done in South Africa, Zimbabwe, and Uganda supports this, stating that although friends and peers were the main source of information on PrEP, it also led to many misconceptions [29]. Participants suggested that qualified HIV care providers conduct widespread campaigns of public education about PrEP to disseminate the essential information about it.

To encourage AGYWs to take PrEP, we discovered that the youth-friendly service center was a platform that could be leveraged. Adolescent females and young women have demonstrated a greater sense of urgency when they visit Youth Friendly Health Service centers, which improve social capital [30]. The youth friendly health services provide confidential and non-judgmental spaces for young people to share concerns, fostering connections, encouraging participation and engagement. In addition, it provides consistent and reliable support to the PrEP users that in turn build self-esteem and confidence among the youth. This is consistent with existing literature which revealed that the youth-friendly clinic was potentially a "low-hanging fruit" for PrEP [30] and that sites that had providers trained in youth-friendly service delivery and offered fully integrated services in private; "youth only" spaces, had better PrEP uptake [30].

The health facility's capacity to take advantage of its current relationships with essential stakeholders was another crucial component of the PrEP implementation's success. This also builds on Chimbindi et. al [31]. and Djomand et. al's [32] findings in the Determined, Resilient, Empowered, AIDS free, Mentored, and Safe women (DREAMS) initiative where they asserted that multisectoral collaborations strengthen existing resources and policies and also promote rapid expansion of the initiative resulting in increased PrEP uptake [33,34]. Djomand et. al. underlined that in order to successfully scale up PrEP services, it is imperative that stakeholder meetings be held with strong, ongoing participation from the MoH, that the government actively oversee its national PrEP program, and that PrEP be promoted outside of the clinical environment [32].

We discovered that the usage of PrEP is hampered by prior experience and predicted side effects. The issues with the short- and long-term safety of PrEP among those at risk have also been noted in previous research from a variety of demographic groupings including MSM, gays, HIV-negative cis women, bisexuals, people who inject drugs and transgender, and our study likewise reported these concerns [35–37]. This reinforces earlier research showing that fear of side effects was a major deterrent to taking PrEP [38].

Our results, which support previous research, indicate that the scarcity of healthcare professionals impedes the delivery of PrEP [39]. It has been noted that introducing PrEP services into a clinic environment that was already beset by a lack of staff and inadequate resources

made matters worse. The healthcare system in Malawi is strained due to the increasing responsibilities of these professionals, who are among the most overworked staff members [40]. Increased workload among HCWs also emerged as a key challenge in the PrEP Implementation for Young Women and Adolescents (PrIYA) study in Kisumu, Kenya [41]. This barrier can be resolved by applying a variety of strategies, including shifting PrEP tasks to lower-level cadres (e.g., peer educators) [42] and the use of pharmacy-based models [43] whereby PrEP clients can get PrEP from community pharmacies. This strategy would increase access to PrEP for people who might not otherwise receive it because of financial or geographical constraints by lowering the number of medical visits required and the accompanying costs. In our study, accessibility concerns were also mentioned as a barrier. This has previously been noted among young individuals in Uganda and a high-risk group of persons in Zimbabwe [29,44]. This further supports the findings of another study in USA that people living in regions with higher PrEP clinic density were significantly more willing to use PrEP [45]. It is therefore recommended to consider the use of strategies that may bring PrEP closer to the clients such as the use of a community pharmacy-based strategy.

The fear of stigma is not new in the context of HIV services [46]. Stigma still prevents adolescent girls and young women from successfully implementing PrEP, even with ongoing community involvement and education. In addition to fearing judgment from friends, family, and sexual partners, participants also anticipated being viewed as promiscuous or HIV-positive if they used PrEP. The work that has been published has defined the stigma associated with PrEP and developed ways to combat it. To date, the most successful methods for reducing stigma have been positive marketing of PrEP and ongoing education at the individual and community levels [47].

Our use of CFIR in this work has shown that we understand the challenges, facilitators, and contextual elements that are specific to PrEP implementation. According to what we've learned, specific recommendations for utilizing the CFIR include applying it in line with the research's implementation phase and integrating it into study design, data gathering, and analysis. An organized, thorough, and prompt knowledge of the obstacles and enablers to PrEP uptake was facilitated by using the CFIR to direct data collection and reporting of findings.

## Study strengths and limitations

Having a large sample size of participants from various backgrounds, including AGYWs using PrEP or not, and HCWs with varying years of service and professional backgrounds, is one of our study's benefits. However, there are certain limits to the interpretation of the findings of this study. First off, social desirability bias was probably present in the participants' accounts of using PrEP. Participants were informed that their answers would be kept private and that there was no right or wrong response in order to lessen the impact of this problem. Finally, since AGYW were specifically from an urban environment in Lilongwe, the results of this study might only apply to AGYW residing in comparable urban environments in the area.

## Conclusion

In conclusion, a complete strategy that is adapted to the specific needs of AGYW, removes obstacles and builds on facilitators, and guarantees access to high-quality services is required to fully realize the potential of PrEP as an effective HIV prevention intervention among them. The key to a successful PrEP implementation is addressing the social and demographic determinants that limit access to and use of PrEP treatments. It also involves strengthening health systems to guarantee the provision of comprehensive and high-quality services. Since the majority of AGYW are now unaware of PrEP, demand generation is essential.

## Supporting information

**S1 Appendix. Interview guide for AGYW-English version.**
(DOCX)

**S2 Appendix. Interview Guide for Health Care Workers in English.**
(DOCX)

## Acknowledgments

The authors would like to appreciate the management of Lilongwe District Hospital, Kawale Health Centre and all study participants for their numerous contributions that made this study a possibility.

## Author contributions

**Conceptualization:** Simon Chichitike Nicholas, Mina C. Hosseinipour, Mitch Matoga, Alinane Linda Nyondo Mipando.

**Data curation:** Simon Chichitike Nicholas, Maureen Matewere, Mercy Tsidya, Mina C. Hosseinipour, Mitch Matoga, Alinane Linda Nyondo Mipando.

**Formal analysis:** Simon Chichitike Nicholas, Agatha Bula, Mercy Tsidya, Alinane Linda Nyondo Mipando.

**Funding acquisition:** Simon Chichitike Nicholas, Mina C. Hosseinipour.

**Investigation:** Simon Chichitike Nicholas, Maureen Matewere, Alinane Linda Nyondo Mipando.

**Methodology:** Simon Chichitike Nicholas, Mitch Matoga, Alinane Linda Nyondo Mipando.

**Project administration:** Simon Chichitike Nicholas.

**Resources:** Simon Chichitike Nicholas.

**Software:** Simon Chichitike Nicholas.

**Supervision:** Alinane Linda Nyondo Mipando.

**Validation:** Simon Chichitike Nicholas, Agatha Bula, Mina C. Hosseinipour, Mitch Matoga, Alinane Linda Nyondo Mipando.

**Visualization:** Simon Chichitike Nicholas, Mercy Tsidya, Mina C. Hosseinipour, Mitch Matoga, Alinane Linda Nyondo Mipando.

**Writing – original draft:** Simon Chichitike Nicholas.

**Writing – review & editing:** Mina C. Hosseinipour, Mitch Matoga, Alinane Linda Nyondo Mipando.

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
