## [Decision Letter · Decision Letter 0]

30 Sep 2024

PGPH-D-24-01884

Barriers and facilitators to Oral Pre-Exposure Prophylaxis Uptake among Adolescents girls and young women at elevated risk of HIV acquisition in Lilongwe, Malawi : A qualitative study.

Dear Dr. Nicholas,

Thank you for submitting your manuscript to PLOS Global Public Health. After careful consideration, we feel that it has merit but does not fully meet PLOS Global Public Health’s publication criteria as it currently stands. Therefore, we invite you to submit a revised version of the manuscript that addresses the points raised during the review process.

We look forward to receiving your revised manuscript.

Kind regards,

Dvora Joseph Davey

Academic Editor

Journal Requirements:

1. Please send a completed 'Competing Interests' statement, including any COIs declared by your co-authors. If you have no competing interests to declare, please state "The authors have declared that no competing interests exist". Otherwise please declare all competing interests beginning with the statement "I have read the journal's policy and the authors of this manuscript have the following competing interests:"

2. Your current Financial Disclosure states, “The author(s) received no specific funding for this work.”. However, your funding information on the submission form indicates that you received funding from “Malawi HIV Implementation Research Scientist (MHIRST) Training Program University" and "NCD BRITE" with Grant Recipient "Mr Simon C Nicholas". Please indicate by return email the full and correct funding information for your study and confirm the order in which funding contributions should appear. Please be sure to indicate whether the funders played any role in the study design, data collection and analysis, decision to publish, or preparation of the manuscript.

3. We note that your Data Availability Statement is currently as follows: "I am sure that all the data used is contained in the manuscript."

4. We do not publish any copyright or trademark symbols that usually accompany proprietary names, eg (R), (C), or TM  (e.g. next to drug or reagent names). Please remove all instances of trademark/copyright symbols throughout the text, including "NVivo® 12.6" on page 6 and 11.

5. We have noticed that you have cited Table 1 in the manuscript file but there are no corresponding tables in the manuscript. Please amend your manuscript to include this table, noting that tables should not be uploaded as individual files.

Additional Editor Comments (if provided):

Reviewers' comments:

Reviewer's Responses to Questions

**Comments to the Author**

1. Does this manuscript meet PLOS Global Public Health’s publication criteria ? Is the manuscript technically sound, and do the data support the conclusions? The manuscript must describe methodologically and ethically rigorous research with conclusions that are appropriately drawn based on the data presented.

Reviewer #1: Yes

Reviewer #2: Partly

Reviewer #3: Partly

2. Has the statistical analysis been performed appropriately and rigorously?

Reviewer #1: Yes

Reviewer #2: N/A

Reviewer #3: Yes

3. Have the authors made all data underlying the findings in their manuscript fully available (please refer to the Data Availability Statement at the start of the manuscript PDF file)?

Reviewer #1: No

Reviewer #2: No

Reviewer #3: Yes

4. Is the manuscript presented in an intelligible fashion and written in standard English?

Reviewer #1: Yes

Reviewer #2: Yes

Reviewer #3: Yes

5. Review Comments to the Author

Reviewer #1: The authors explored the factors that facilitate PrEP uptake and barriers to the same among adolescent girls and young women at increased risk of acquiring HIV infections.

I believe this is a significant study due to the low uptake of PrEP despite the demonstrated efficacy of the ARVs in preventing HIV infection.

Strengths: Including participants not on PrEP in the study provided an avenue to explore barriers affecting PrEP uptake

Major Issues

1.Introduction: reference number 16 which should have provided details on the targets for PrEP was not found. Authors should investigate that. Authors should provide additional details on the PrEP cascade including how many AGYW that received HIV testing services tested negative, the number offered PrEP, and the number that accepted PrEP and enrolled.

“This study deliberately chose non-probability sampling [19] to include both AGYW who accepted PrEP and those that were not on PrEP (10 AGYW in each group)”

2.Sampling: Mention was made of appendices 1 & 2 which contain the interview guides and vignettes but are not included in the paper. Is this a deliberate omission?

3.Results: Authors to clarify if side effects were barriers to PrEP uptake or PrEP continuation. Is it fear of side effects or presence of side effects?

Minor Issues

1.Sampling: Study Authors should clarify the categories of participants recruited into the study. Those not on PrEP, were they offered PrEP and found to be eligible for PrEP but declined?

Reviewer #2: Thanks very much to the authors for the opportunity to read this manuscript. Overall, the topic is important, and with greater detail in the results section and improved connection between the framework used to design the study and the discussion section, it will be a nice contribution to the literature. I would recommend a review for clarity of language and grammatical issues throughout prior to resubmission.

Abstract:

In the results section of the abstract, there are a lot of different themes summarized in two long sentences. It would help if the themes for AGYW were discussed in separate sentences from those for HCW and more detail included. For example, it is said that HCW attitudes (there's a typo - it says altitude but that should read attitudes) are facilitators, but I could see those potentially being barriers depending on what the attitude specifically is. More detail should be included to clarify.

In the conclusion of the abstract, the statements made are not clearly built upon the results highlighted in the previous section. For example, what is the comprehensive strategy that is needed that is tailored?

Introduction:

Paragraph 2: In the first line it says that the disparities in the impact of HIV are attributed to high risk behavior. Do you mean among AG YW or more broadly among youth? Space in addition you say that place of residence has also been associated with risk of HIV. Clarifying which places have an increased risk would be helpful for readers unfamiliar with this topic. In the sentence that begins with “despite availability of female condoms...,” there is a typo with the word usage. Should that be use? In addition, would it be helpful here to explain the reason why women have limited ability to use or negotiate economies? For example, gender inequitable norms that influence dynamics?

I would also recommend starting a new paragraph before the sentence that begins “another HIV prevention intervention is...”

Paragraph 3: if there has been limited research detailing the factors that influence PrEP uptake, what has that limited research shown? If there has been no research to date, then it should be explicitly stated. Since the study setting is an urban area it might be helpful to spotlight findings of prep uptake in urban versus rural areas. Is it higher in urban areas in Malawi or not, for example? That would help support your study justification a bit more.

Methods:

Study setting: in the sentence starting “The services offered…,” there is a missing “are” after the word include.

Sampling: while you used purposive sampling, it would be helpful to give more details about how you recruited participants. For example, did it take place in person, using flyers, or another method?

Data collection: What do you mean by “well-trained?” what training was provided?

Why was the CFIR framework used to guide the interview guide? How? Can you elaborate a bit more?

Data analysis: Change second sentence to “Data “were” managed…” In addition, the authors jump from familiarization with transcripts to talking about how codes are compared, but there should be a bit more discussion of how the codebook was developed and tested. How can codes be compared before developing the coding framework? How did the authors use the CFIR framework to guide coding? Once coding was complete, how did the authors identify salient themes? What process was used?

Results:

In the demographic characteristics section, it might be helpful to clarify what the different HCWs’ roles are in particular related to PrEP administration.

In table 1 under Characteristics of Individuals, Facilitators – the summary is described as “knowledge of PrEP,” but what is written extends beyond PrEP to include knowledge of HIV risk susceptibility as well. In the Inner Setting section, under burden of work, it would be helpful to have the authors elaborate more on why high workload was considered a barrier to PrEP uptake. In addition, some of the factors mentioned seem like they might fit better in the “outer setting” section. Perhaps why the authors have bucketed them in “inner setting” could be clarified. In the outer setting section, the patient needs and resources under facilitators is not clear.

Facilitators: knowledge – the first sentence is quite long, with some grammatical issues, and should be shortened and made into multiple sentences to explain the findings in greater depth.

In the translated quotes throughout the results, I would suggest minor tweaks to the grammar to retain the original meaning but make them more easily understood by the reader. In addition, it would be nice to integrate explanation or introductions to each quote, rather than having them included at the end of each section without context.

Facilitators: inner setting – the availability of resources section is quite short and could have greater detail. In addition, I wonder if it makes sense for this to be in the “inner setting” section or the outer setting?

Barriers: Adherence – The point about abuse of PrEP could be clarified further to illustrate the point being made.

Barriers: workload – It would be helpful to expand the theme around workload to clarify why exactly this is a barrier for PrEP uptake. Similarly, when talking about limited resources, it is mentioned that they are unable to conduct some of the critical tests. Can the authors incorporate an additional discussion around how this then prevents people from using PrEP? The point is implied, but I think it can be made clearer by the authors.

Barriers: outer setting – It would be nice to incorporate more detailed discussion of the theme around stigma, as this is an important emergent finding to highlight, particularly in terms of programs working to improve uptake.

Discussion:

It is important that the authors highlighted some of the findings around those with sex work experience. How many participants had this experience? How did the authors ensure saturation was reached with that theme as it was not part of the sampling strategy?

Paragraph beginning with “In order to…” – suggest removing “really” in the first sentence to make the writing more formal. In addition, I am not sure I understand how visiting youth friendly services improves social capital. This point would be stronger if elaborated on in greater depth.

Paragraph beginning “We discovered…” – It would be helpful to repeat the specific demographic groupings here to inform policy and program thinking.

It would be helpful to explain the implications of the point around stigma and perceived risk compensation made in the discussion. Do the authors see these as connected? What are the programmatic and policy implications of this? How does this link back to the findings in the results?

In addition, it might be helpful to circle back to the CFIR framework in the discussion, given its important role in the study design and analysis. What are the implications of these findings informed by the CFIR?

Reviewer #3: METHODS

Study setting

STI cannot be described as a service. Services may be either prevention of STI or treatment of STI or both. Authors should correct this.

Authors also did not mention maybe the health facility offers provision of PrEP as one of the services.

Sampling and recruitment of study participants

Authors may need to clarify if the 10 AGYW not on PrEP selected for the survey were not eligible for PrEP, or eligible and not offered or were offered but declined. This may be helpful to know as it may influence their responses.

Were the 10 HCWs selected the ones trained on PrEP or a mixture of those trained and those not trained?

Data collection

Authors may need to clarify how the interviews were carried out……the interview of the AGYW seems like a form of exit interview, that is they were interviewed after accessing services and leaving the facility. The interview was said to be carried out within 28 days, so how many respondents were interviewed per day, did all the required number of respondents (20) show up within that 28 days for services, or some needed to be called to come for the interview? Maybe additional information about AGYW client volume in Kawale health center and the AGYW PrEP clients’ visit/appointment frequency may be helpful to understand how the 20 respondents were reached in 28 days via exit interview method.

Another important part that may require clarity are the data collection instruments used, it was indicated that the interview guides and vignettes are attached as appendices but they were not seen. An in-depth interview in a qualitative survey is meant to explore respondents’ insight as much as possible, so the instrument used may be better if not structured.

RESULTS

Beyond the feedback from the HCWs, it may be nice to get/document the perspective of the AGYW concerning the attitude of HCWs, and the issue of lack of privacy in the facility. The perspective of the AGYW with regards to these factors will be very helpful to guide our recommendation because they are the recipients of the services provided by the HCWs and the facility.

The perspective of the HCWs concerning the issue of stigma (outer setting factor) may also be documented from this survey if available. As mentioned in the discussion section where some HCWs are reluctant in offering PrEP to AGWYs because of the concern of increase inappropriate sexual behavior and possible ARV drug resistance among the AGYW, are there such among the HCWs interviewed in the Malawi study too, especially if such HCW has not been trained on PrEP provision?

Detailed reviewer's feedback has been uploaded as an attachment

6. PLOS authors have the option to publish the peer review history of their article (what does this mean? ). If published, this will include your full peer review and any attached files.

**Do you want your identity to be public for this peer review?** For information about this choice, including consent withdrawal, please see our Privacy Policy .

Reviewer #1: No

Reviewer #2: No

Reviewer #3: No

---

## [Decision Letter · Decision Letter 1]

26 Feb 2025

Barriers and facilitators to Oral Pre-Exposure Prophylaxis Uptake among Adolescents girls and young women at elevated risk of HIV acquisition in Lilongwe, Malawi: A qualitative study.

PGPH-D-24-01884R1

Dear Mr Nicholas,

We are pleased to inform you that your manuscript 'Barriers and facilitators to Oral Pre-Exposure Prophylaxis Uptake among Adolescents girls and young women at elevated risk of HIV acquisition in Lilongwe, Malawi: A qualitative study.' has been provisionally accepted for publication in PLOS Global Public Health.

Best regards,

Dvora Joseph Davey

Academic Editor